# Appia: Simpler chromatography analysis and visualization

**Richard Posert[1,2], Isabelle Baconguis[2]***

**1** Department of Chemical Physiology and Biochemistry, Oregon Health & Science University, Portland, Oregon, United States of America, **2** Vollum Institute, Oregon Health & Science University, Portland, Oregon, United States of America

* bacongui@ohsu.edu

## Abstract

Chromatography is an essential family of assays for molecular biology and chemistry. Typically, only a qualitative assessment of peak height, position, and shape are sufficient to proceed. Additionally, chromatography instrument software is proprietary and often locked to a single computer, making data analysis and sharing difficult. Since each manufacturer reports the data in their own proprietary format, performing analysis of experiments which use multiple instruments or sharing data between labs is also challenging. Here we present Appia, a free, open-source chromatography processing and visualization package focused on making analysis, collaboration, and publication quick and easy.

## Introduction

Chromatography is a fundamental assay for investigating the quantity, homogeneity, and purity of small molecule and protein samples. In its most basic form, chromatography involves measuring the spatial separation of an analyte induced by flowing a mobile phase over a stationary matrix [1]. While the mobile phase flows through the matrix, sample is carried along at a varying rate based on the property to be analyzed. For example, in size-exclusion chromatography (SEC), the smaller analytes move more slowly through the packed resin than larger ones, while in reverse-phase chromatography peak position correlates with the relative hydrophilicity of the analyte [2, 3]. Thus, peak position (that is, the time or volume at which sample elutes) informs the user about the relative affinity of the analyte for the matrix (corresponding to size or hydrophobicity in the examples given). Peak shape (the width and height of the sample peak) gives information about the homogeneity of the species with regard to the measured property. Peak area gives the total amount of analyte injected, but peak height is often used as a heuristic for peak area (which can be difficult to accurately measure). For most experiments, the peak position, height, and width information about the analyte are sufficient for proceeding with the next experiment.

The simplicity of these three main readouts (the position, height, and width) means that most day-to-day chromatography experiments can be satisfactorily analyzed by eye in a few seconds. Indeed, screening data from a type of SEC ubiquitous in the field of structural biochemistry are often laid out in a large grid of very small plots because sample quality is so

**Funding:** This work was supported by the National Institute of General Medical Sciences (T32GM071338, initially to support RP, and

R01GM138862 to IB). The content is solely the responsibility of the authors and does not necessarily represent the official views of the National Institutes of Health. The funders had no role in study design, data collection and analysis, decision to publish, or preparation of the manuscript.

**Competing interests:** The authors have declared that no competing interests exist.

immediately recognizable [4]. This layout hints at an issue—simple assays require simple presentation to be legible. However, such presentation is incompatible with complex methods which might be performed on the same instruments. Thus, software written by chromatographic instrument manufacturers must be more complex and, unfortunately, more difficult to use. Moreover, each chromatographic instrument manufacturer develops their own analysis software, meaning that data formats are not inter-operable, and often users must switch between two or more different programs to analyze a single experiment. This appears a simple inconvenience, but there is a great body of research indicating that interface complexity and "task-switching" (in this case, switching between two different manufacturer software packages) wastes effort and makes learning and insight more difficult [5–7]. Given how common chromatography is, even small wastes of time or energy add up to a significant delay over the length of a project. We wrote Appia to solve these problems by giving simple assays a simple interface.

Appia is a free, open-source project which comprises a set of data processing scripts and an optional, locally hosted web service (Fig 1). The entire project is free and open-source. The processing scripts convert data from proprietary manufacturer formats to portable.csv files, and the web service allows users to access their data from any computer with access to the server. The web service is hosted locally, meaning that users maintain full control over who does and does not have access to their chromatography data.

## Results

First, we will describe the actual functionality that Appia provides. Next, we will demonstrate its utility in day-to-day chromatography experiments, with a focus on protein biochemistry.

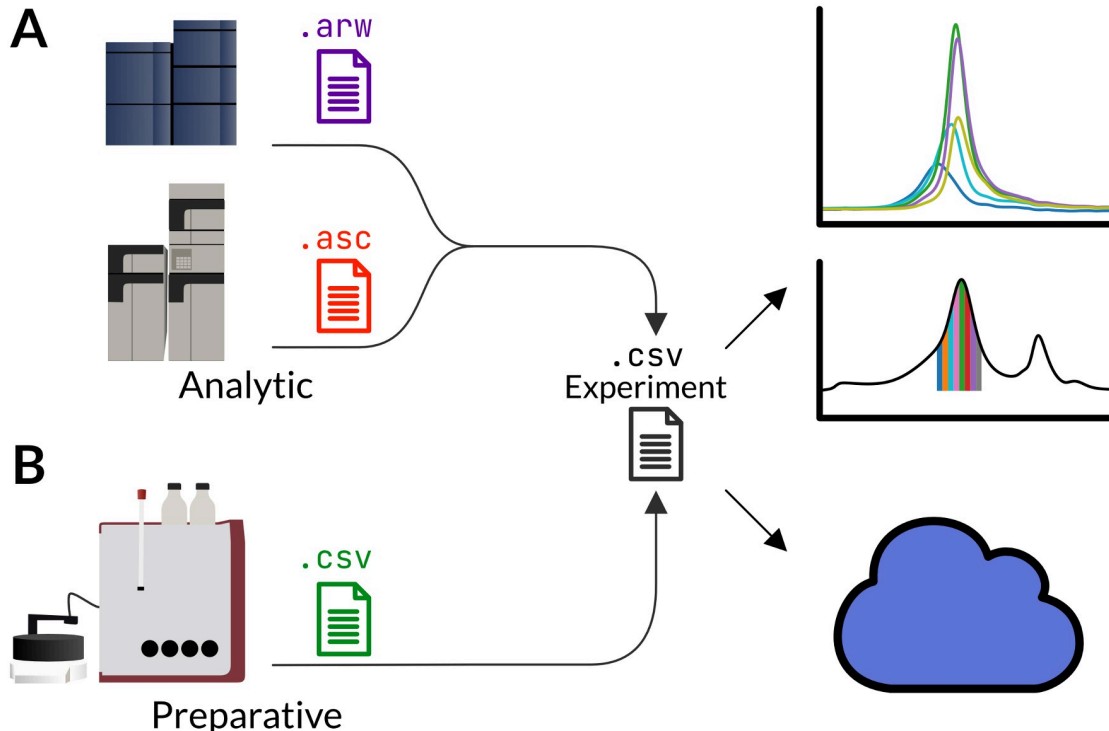

**Fig 1. Appia overview.** Appia processes proprietary data formats (colored) from a variety of manufacturers and automatically determines whether the data are analytic (A) or preparative (B). Appia saves this data in standardized CSV format files (black) for plotting. Chromatography data are uploaded to a local Appia Web database at the same time.

We focus on protein biochemistry because that is the main focus of our lab. However, Appia can perform the same process on any supported chromatogram's data.

## Appia processing pipeline

Before they can be visualized, chromatography data is processed into an Appia "experiment". The user provides raw data files exported from their chromatography instrument. Appia will prompt the user for basic information about the run if that information is not included in the data files. The minimum required information to create an Appia experiment is the flow rate and channel name. Some chromatography manufacturers provide this information in the file, in which case it is read automatically and no user input is needed. If the information is not available in the file, users can provide this information individually for each trace, together for a set of traces processed in the same batch, or one time in Appia's internal database. Users can also choose to provide a scaling factor for the entire batch of processed traces. This could be used to account for differing flow cell path lengths between instruments. Other options available at processing time include normalization range, automatic plot generation, experiment naming and combination, and others.

Next, Appia combines this data into a standard "tidy" data format which is written to a CSV file. Appia can currently convert data from all major chromatography instrument manufacturers' (Waters, Agilent, Shimadzu, and GE/Cytiva) proprietary format into a portable, user-friendly format [8]. The tidy data format is a standard in the fields of data science and visualization, in which each row of the table represents a single "observation" (perhaps, the absorbance of a single sample at a single time point for some single wavelength), while the wide format stores some of this information in the column headers (Fig 2). Both forms of the data are made available to users should they want to plot data using other program or package, such as ggplot2 or matplotlib (which both use the long, tidy form), Prism or Excel (which both use the wide form).

After data are processed, they can be visualized. The user can perform their own visualizations for more advanced experiments using the CSV file, but Appia provides functionality to automatically create appealing basic plots during processing. Additionally, if users set up an Appia Web server, the data will be uploaded to the Appia Web database for viewing through a browser. The database is locally hosted—we (the authors) neither host nor have access to any data processed by other Appia installations. The database is locked behind a username and password which are required for upload and download of data. The web interface is by default open to anyone with knowledge of where it is hosted but can easily be served behind a username and password using NGINX or some other server tool. Since Appia Web is installed as a docker container, if users wanted their own unique Appia database, several separate installations could be hosted on the same machine with different usernames and passwords.

## Appia Web

The Appia Web interface provides a centralized location for viewing all experiments processed with Appia (S1 Video). A demo of Appia Web is available at http://traces.baconguislab.com. Users can scroll through an experiment list, or type to search. Once selected, the data are displayed in zoomable, interactive line plots. Data are separated into analytic data, since these experiments typically require different visualization. For preparative data, the overall trace is displayed with fractions highlighted in color below it. To ease comparison, preparative traces can be overlaid on the analytic data when desired. Analytic data is displayed with a single line per injection, colored by injection name. Appia can handle any number of simultaneous acquisition channels, which are all plotted separately. Users can click the color legend to turn on

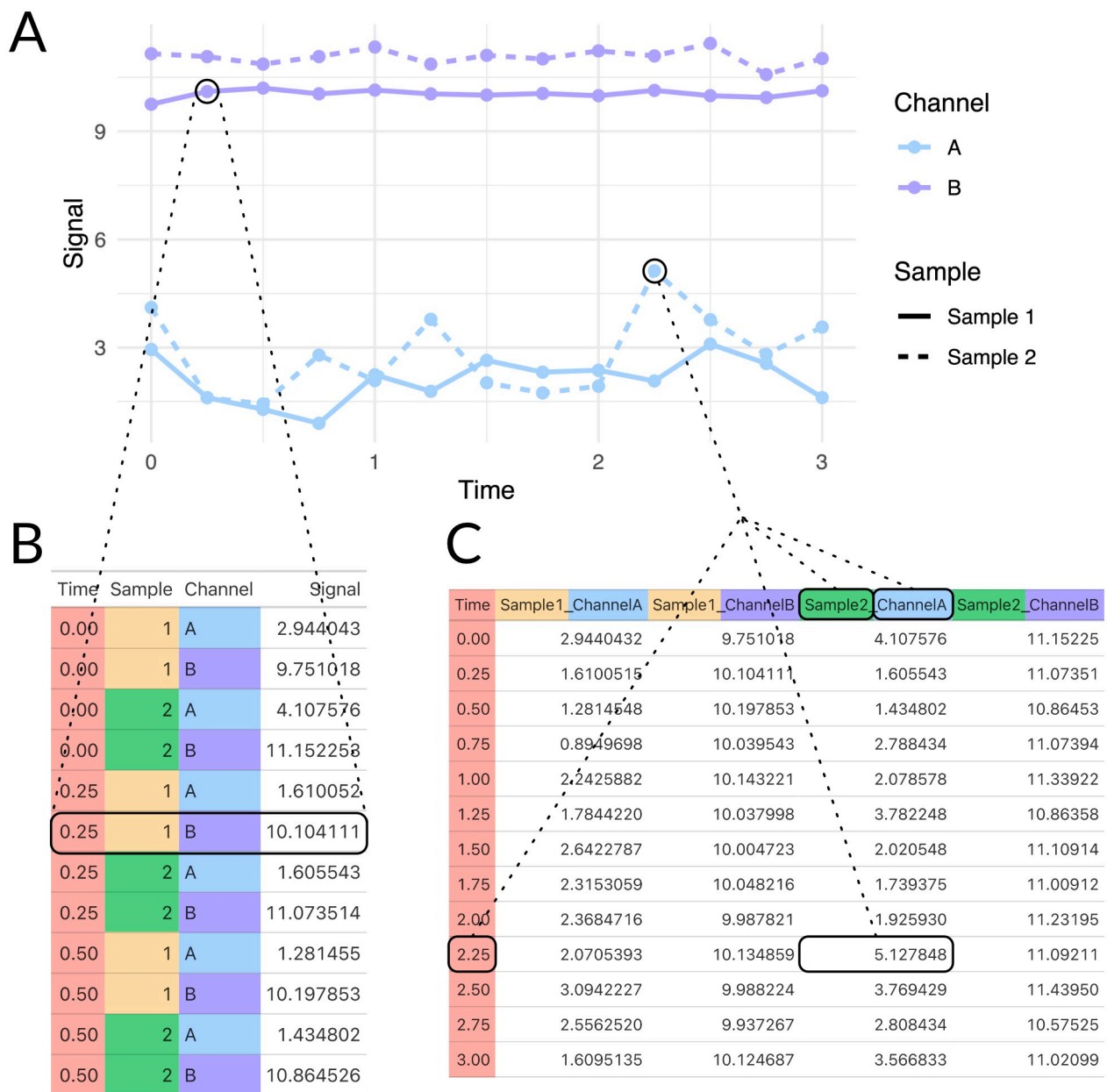

**Fig 2. Appia data formats.** A: Example data plot of generated data. B: The data plotted in A, represented in "tidy" format. Each column represents a variable, meaning each row represents a single observation (or a point in A). C: Wide format. One column represents a variable, and the other columns represent groups of observations along that first variable. This means that variables beyond the first must all be stored in the column headers, so information about a single observation is spread throughout the table, but the table is more compact. Both "tidy" and wide data are produced by Appia.

and off display of fractions or traces. Additionally, users can zoom in to a region of the trace and normalize the maximum value of each injection in that region to 1. Normalization occurs internally to a single trace, i.e., the signal range for each injection/channel pair over the selected interval is linearly transformed such that the maximum value is 1.0 (Fig 3). During processing, the user may optionally select to also have the minimum value set to 0 (i.e., set the total range of each trace to [0,1]) to correct simple baseline issues. These normalized plots,

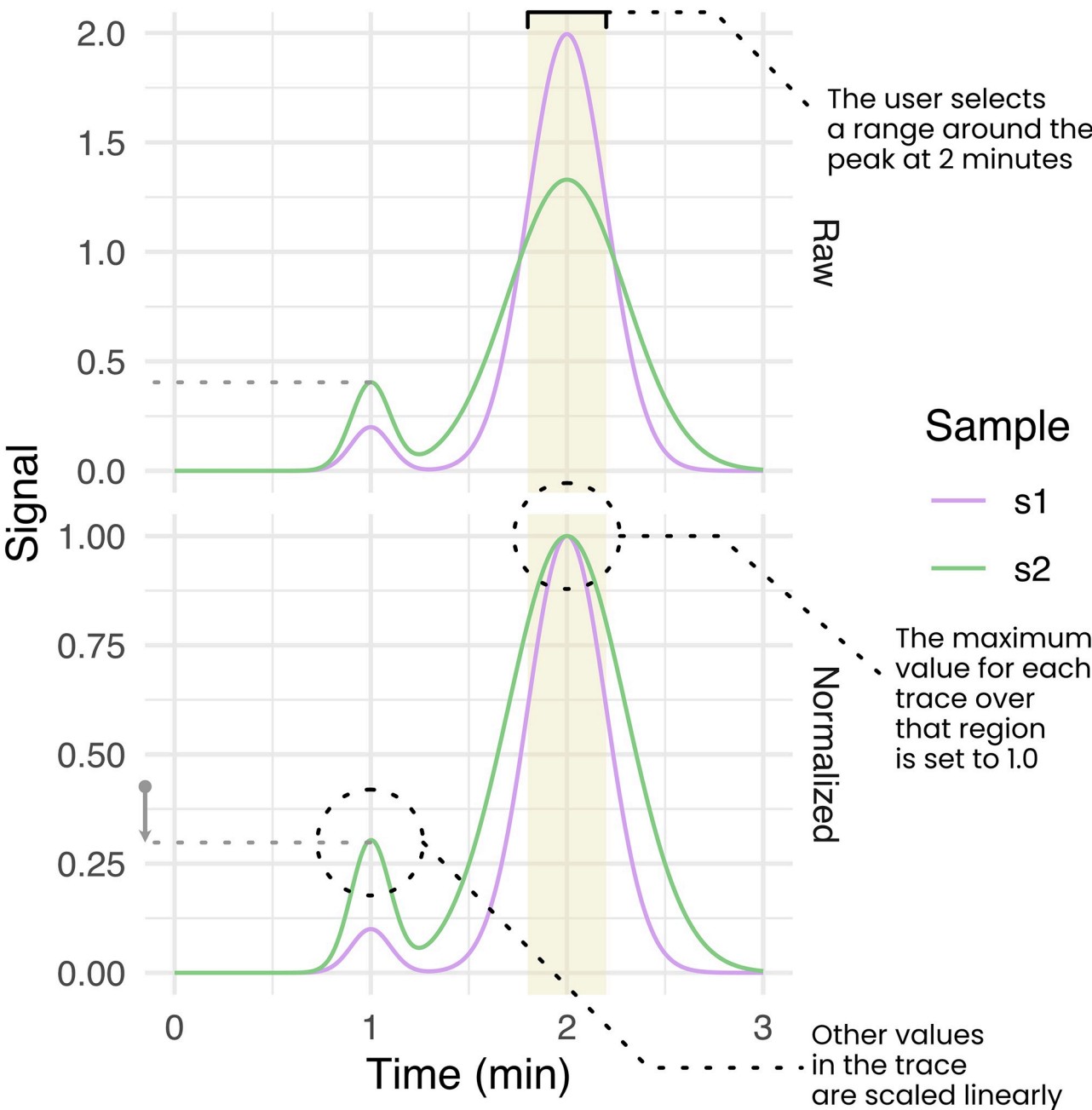

**Fig 3. The Appia normalization process.** Users first select a range over which to normalize, either by zooming into it in the web interface or entering it in the command line during processing. For each trace, the maximum value over that range is set to 1.0, and all other values along the trace are scaled linearly. Optionally, during processing (but not the web interface), the minimum value can be set to 0 as well.

presented alongside the raw-data plots, make it easy to compare peaks from different injections to a reference peak. Once a satisfactory image is created, it can be downloaded directly from the web interface, as can data files containing both normalized and un-normalized values.

If a user selects multiple experiments, they are combined. For analytic data, injections are labeled with their respective experiment names and plotted together for ease of comparison.

Preparative data for combined experiments show the overall injection profile colored by experiment without fraction fills. As with single experiment display, all graphs are fully interactive. The list of selected experiments, X-coordinate zoom, and normalization region are continuously stored and updated in the URL query string. This means that a link shares not only the list of experiments, but a particular region of the trace to be inspected (for example: http://traces.baconguislab.com/HPLC_Example_1+HPLC_Example_2?view-range=1.93-2.13).

## Examples of use

All of the above make chromatography data easier to work with, but thinking of it as merely a tool of convenience ignores difficulty of gaining insight from data which are difficult to work with. For instance, a common assay in our lab is exploration of expression levels of a given protein construct fused to a fluorescent protein, such as mKalama, mVenus, or tdTomato. We infect cells with a virus, causing them to express the construct of interest. We can then track expression levels and species homogeneity by tracking the fused fluorescent protein. The ideal expression level is often a function of multiplicity of infection (i.e., the ratio of virions to host cells; MOI), additive concentration, expression time, and temperature. Appia's standard data format means that users can write a processing script or develop a Prism/Excel workflow once, and then apply it quickly to the data every time this assay is run (Fig 4A).

Manufacturer software, on the other hand, displays all the lines in the same pane (Fig 4B); users must remember which line corresponds to which set of conditions. In this case, it is very difficult to follow the decreasing maximum protein expression with increasing MOI, and even more difficult to notice that the duration effect disappears with higher MOI. For this

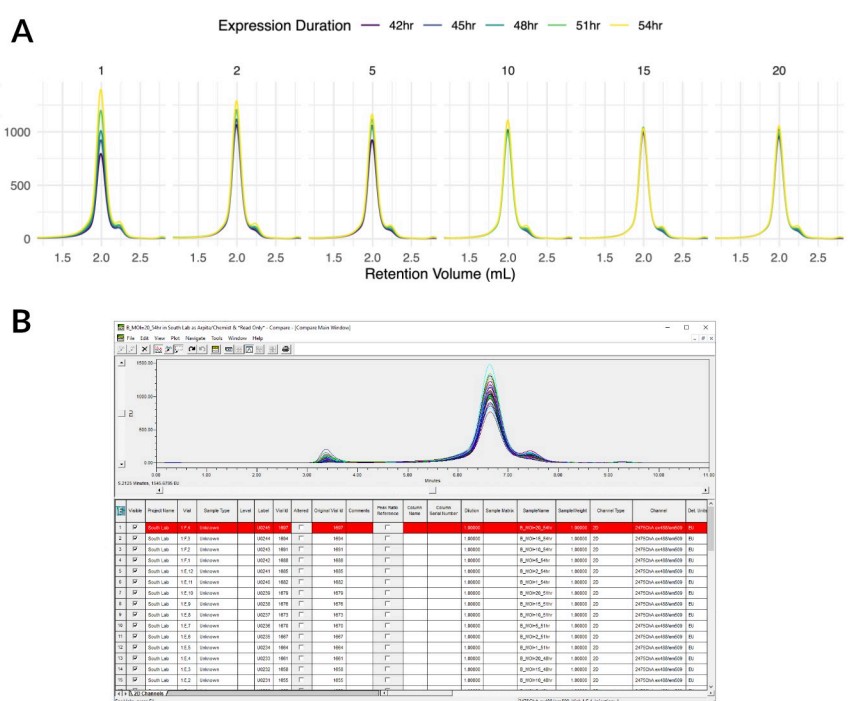

**Fig 4. Comparison of Appia and manufacturer software.** Small cell pellets with the indicated baculovirus MOI and expression duration were thawed, solubilized in DDM, and run in a Waters Acquity Arc Bio HPLC over a Superose 6 Increase 5/150 column. A: Data processed and visualized with Appia and ggplot. Data export and visualization took approximately 20 seconds. Trends in both expression duration and MOI are obvious, and the user can move on with larger-scale expression tests. B: The same data as viewed in the manufacturer software.

experiment, we selected an MOI of 2 and 42 hours, even though the maximum is at MOI 1 and 54 hours, as the slight gain from longer expression is not worth the time. While this classic analysis has been performed in our lab for years, it now takes a fraction of the time thanks to Appia's clear plot layout. Additionally, Appia's standard data format means that even if a similar experiment were run on a separate instrument from a different manufacturer, the same visualization and processing scripts could be used, eliminating the wasted effort of learning a new manufacturer software. Although we present only protein data here, our collaborators have used Appia to automate analysis of the stability of a compound. They monitor absorbance peak height and use an automated plotting script to quickly compare the peak of interest with coloration by time (data not shown). This use case demonstrates the power of Appia's general applicability and standard data formatting.

Appia centralizes data analysis workflows. For instance, our lab relies extensively on paired Fluorescence-detection Size Exclusion Chromatography (FSEC)/SEC assays. First, several constructs of interest are purified in a matrix of conditions. These lysates are assayed for homogeneity and yield by FSEC. All the resulting plots can be viewed simultaneously, from anywhere with an internet connection, in Appia Web (Fig 5A). We next express a larger quantity of the most promising condition and purify it by affinity chromatography and SEC. Again using Appia Web, we select the most promising fractions of the preparatory SEC (Fig 5B) and assess them again for monodispersity using FSEC (Fig 5C). We can compare the peak profiles directly with previous purifications of the same or different conditions and constructs, or our small-scale tests, by loading up the relevant experiments in Appia Web. The single-destination simultaneous display of both preparative and analytic data makes comparison and selection both easier and faster, and simultaneous display of raw and normalized data allows for direct comparison of both yield and monodispersity without any need for user input or task switching.

Appia Web also makes binding interactions easier to analyze (Fig 6). The user can normalize FSEC traces to the unbound peak. This makes the relative peak heights a proxy for the relative proportion of the fluorescent species in the bound or unbound states. For instance, here we show a binding interaction between the triheteromeric αβγ Epithelial Sodium Channel (ENaC αβγ) and its regulatory partner NEDD4-2, tagged with GFP [9]. ENaC αβγ is untagged, so any GFP signal indicates the position of some species containing NEDD4-2. It is plainly clear from the raw traces that there exists some binding interaction between the two, and the traces normalized to unbound peak height give a heuristic approximation for relative binding: at a 1:5 NEDD4-2:ENaC αβγ ratio, the unbound peak is about 4 times higher than the bound peak, giving an approximate 20% of the total NEDD4-2 population bound by ENaC αβγ.

## Discussion

Chromatography is a workhorse assay of chemistry and molecular biology. Because of the assay's flexibility, manufacturer software is often difficult to use for simpler, routine assays, because it must present functionality required for advanced chromatographic assays. Appia simplifies analysis of experiments where only the simple readouts of peak shape, heigh, and position are required by producing easy-to-read plots, giving users insight into their data with little distraction.

Appia's standard data formats and web interface also make data sharing very simple. Collaborators can directly view the chromatogram of purified material from their own computer and download the data for further investigation if needed. Even if two labs use different chromatography instruments, Appia's standard data format means that there are no extra efforts spent learning a new set of software to analyze the relevant data. Moreover, Appia Web

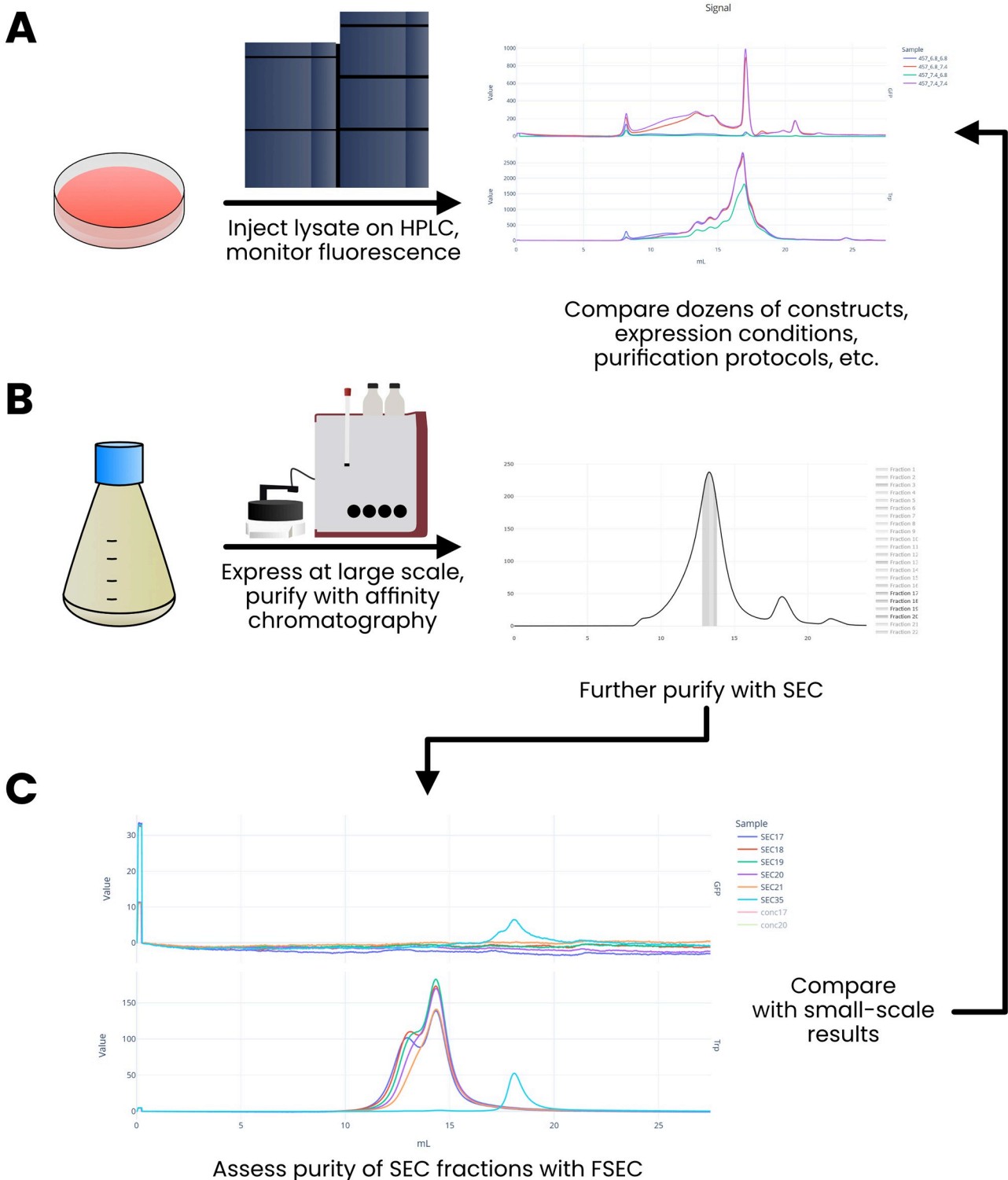

**Fig 5. Paired SEC/FSEC.** In a paired SEC/FSEC assay, small-scale purifications of varying constructs in varying conditions are analyzed by FSEC for monodispersity and yield (A). Next, conditions of interest are purified by affinity chromatography and preparative SEC (B). Fractions from each step are selected with Appia Web for re-analysis using analytic FSEC (C) to assess the homogeneity of the samples. These results can easily be compared with prior analytic work of other constructs or conditions without ever leaving the Appia Web interface.

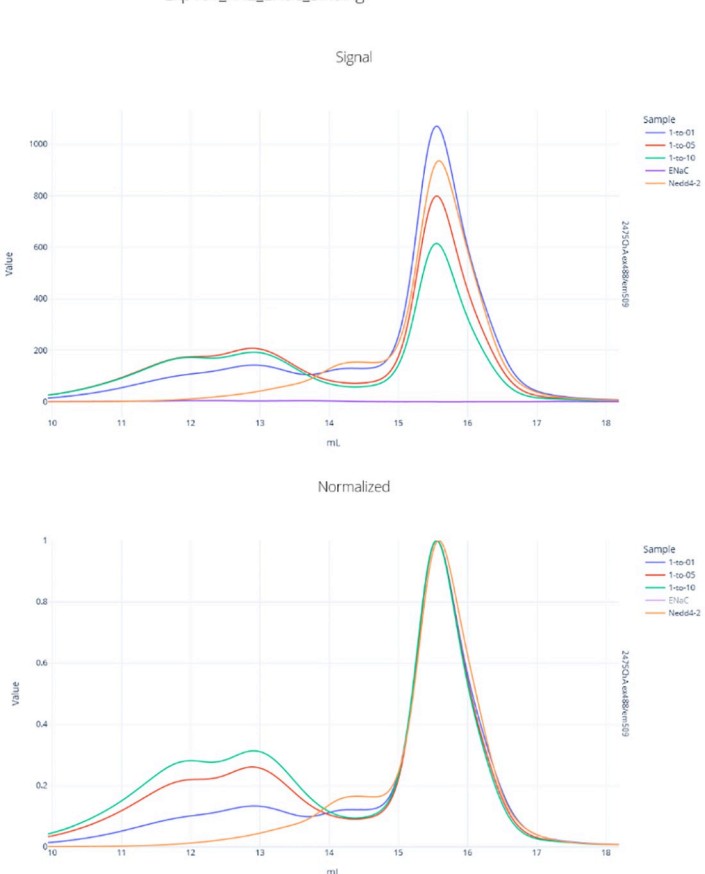

**Fig 6. ENaC αβγ/NEDD4-2 binding.** Top: raw GFP fluorescence from NEDD4-2-GFP fusion protein. Bottom: fluorescence signal normalized, setting free NEDD4-2-GFP peak height to 1.

presents simple one-click controls for common data manipulations such as scaling and peak height normalization, meaning that even advanced chromatography users performing standard assays will find Appia easier and faster to use than manufacturer software.

Appia represents a significant improvement over the current state-of-the-art for chromatography processing and presentation. Currently, most chromatography results are analyzed using the manufacturer software. Custom plots are only produced for publication, typically by manually copy-pasting data from manufacturer outputs to an Excel or Prism file. This process is slow, laborious, and error prone, leading users to prefer analysis directly in the manufacturer software. The layout and functionality of most manufacturer software is not designed to handle multi-variable plots and comparisons between more than a handful of traces. This makes understanding data and seeing trends difficult. Moreover, if experiments require the use of two or more different machines, manufacturer software makes it impossible to view the data all in one interface. Appia, on the other hand, produces simple plots and clean data in a single click. Not only does this reduce barriers to inspection of and insight into the data, but also to sharing data with colleagues. Sending a link to the web interface allows collaborators to quickly compare data with other experiments they've run, or investigate the traces in detail, both of which are difficult if not impossible if a static image or raw data is sent.

Appia is, of course, not sufficient to run an instrument or process advanced chromatography workflows. Instead, Appia is meant to provide a free, open-source interface to make all

parts of day-to-day chromatography simpler: analysis with the streamlined, appealing interface; sharing through Appia Web; and publication through Appia's plotting functionality and standardized data format.

## Conclusion

We have presented Appia, a free, open-source chromatography data processing aid visualization suite. Using Appia gets users from raw data to biological or chemical insight faster and with less effort than using manufacturer software and other plotting programs. Appia Web allows users to inspect their data without tying up the instrument computer and allows for faster and more in-depth collaboration on projects relying on chromatography. Our focus is on simple, heuristic readouts of as many chromatography instruments as possible. We thus plan to focus on widening our supported array of instruments rather than adding more complex features like add support for peak calling and fitting or baseline subtraction beyond the simple normalization currently provided. We do, of course, welcome collaboration on this front.

Appia is under active development and is the main mechanism by which our lab analyzes chromatography data, meaning it will remain usable for years to come. We are happy to take requests to support additional manufacturers and have created a specific section of the GitHub repository to support these requests. Appia is also open-source and written to be easily extensible, so that functionality or additional manufacturer support can be added by users if they desire. It is our hope that wide-spread adoption of Appia makes day-to-day chromatography easier well into the future.

## Experimental procedures

### Appia

Appia is currently written to analyze data from Waters, Shimadzu, and Agilent HPLCs and GE/Cytiva FPLCs, and is under active development. Support of a new chromatography manufacturer requires writing a parser to convert the manufacturer export format to the Appia format; this typically takes between one and two hours for someone moderately experienced in python, and only has to be done once per manufacturer. Requests for additional manufacturer support are very welcome and require only submission of some basic information as well as a few representative chromatogram export files to our GitHub repository. The Appia processing scripts can be installed with a single command using pip ('python3 -m pip install appia') on any machine with a python3 installation (available for free at python.org) and internet access. Appia includes extensive feature documentation in the '—help'functions (e.g., 'appia process–help'for a list of options during processing).

Appia Web is optional, but highly recommended. The web interface requires installation of both Appia Web and an Apache CouchDB database, along with basic networking. Since the networking and installation of these features requires some technical expertise, we provide and maintain a Docker Compose YML template. Docker Compose automatically builds both the Appia Web UI and the CouchDB database and appropriately networks them together with a single command ('docker-compose up'). Once the Appia container is running, we recommend that it is served using NGINX rather than directly accessing the Appia port. This setup enables a username/password login to the web interface. The GitHub repository provides an example NGINX configuration for serving Appia in this fashion. A typical Appia installation would include an installation of python3 and Appia on each instrument computer (to process and upload data) and some central, web-accessible computer running Appia Web. This central web-accessible computer could be one of the instrument computers or a distinct machine.

Appia Web's resource demands are quite low, and we have successfully hosted it on an inexpensive Raspberry Pi. Each user's personal computer does not require any additional software to access Appia Web.

## Expression profiling

HEK293S GnTI⁻ cells were infected with baculovirus for GFP-tagged CMP-sialic acid transporter at multiplicity of infection (MOI) of 1, 2, 5, 10, 15, or 20 [10]. Cells were incubated at 30˚C for 54 hours, with 2 mL aliquots removed at 42, 45, 48, 51, and 54 hours of infection. Aliquots were centrifuged at 3,000 xg for 10 minutes, after which cell pellets were flash-frozen in liquid nitrogen and held at -80˚C until all pellets were prepared. Cell pellets were solubilized in solubilization buffer (50 mM HEPES pH 7.5, 150 mM NaCl, 0.01 mg/ml deoxyribonuclease I, 0.7 μg/ml pepstatin, 1 μg/ml leupeptin, 1 μg/ml aprotinin, 1 mM benzamidine, 0.5 mM phenylmethylsulfonyl fluoride, and 2% (w/v) n-dodecyl-β-D-maltopyranoside [DDM]) for 1 hour at 4˚C, then centrifuged at 35,000 xg for 30 minutes. Next, 50 μL supernatant was injected on a Superose 6 Increase 5/150 column (Cytiva) on a Waters Acquity Arc Bio HPLC and monitored for GFP fluorescence. Data were processed through Appia, then through a standard plotting script developed previously for this type of assay (S1 File).

## SEC fraction analysis

Wild-type-like ENaC δβγ was expressed in HEK293S GnTI⁻ cells and purified as described previously for αβγ ENaC [11]. Briefly, cells were infected with baculovirus at an MOI of approximately 2 and incubated at 30˚C for 72 hours, then harvested by centrifugation and flash-frozen. ENaC-δβγ-GFP cell pellets were solubilized in solubilization buffer (20 mM HEPES pH 7.4, 150 mM NaCl, 2 mM MgCl$_2$, 25 U/mL Pierce universal nuclease, protease inhibitors, and 2% digitonin) for 1 hr at 4˚C, then centrifuged at 100,000 xg for 45 minutes. Supernatant was passed over a GFP nanobody resin to bind ENaC-δβγ-GFP. ENaC was eluted from the resin by treatment with trypsin (33 μg/mL resin) to cleave the channel from GFP. Trypsin-eluted fractions were concentrated and injected on a Superose 6 Increase 10/300 column (Cytiva) on an AKTA Pure FPLC. Fractions were collected and re-analyzed by FSEC on a Waters Acquity Arc Bio. Running buffer was the same for SEC and FSEC: 0.5 mM GDN, 20 mM HEPES pH 7.4, 200 mM NaCl. SEC data and fraction FSEC data were exported and processed with Appia and analyzed entirely in Appia Web.

## NEDD4-2/ENaC αβγ binding

Human NEDD4-2-GFP was expressed in Sf9 cells at an MOI of 5 for 48 hours, then flash frozen. Cell pellets were resuspended in 100 mL TBS (20 mM Tris pH 7.5, 200 mM NaCl) with 25 U/mL Pierce universal nuclease and protease inhibitors per 800 mL cell culture, then sonicated as follows: 5 minutes total sonication, 10 seconds on, 30 seconds off, power level 7. The lysate was centrifuged at 45,000 xg for 45 minutes, then bound to 10 mL TALON resin per 800 mL cell culture. The resin was washed with 5 CV TBS, then 2 CV TBS + 10 mM imidazole, and then NEDD4-2-GFP was eluted with TBS + 250 mM imidazole. NEDD4-2-GFP was concentrated and further purified by SEC. ENaC αβγ is a different triheteromeric channel in the same family as ENaC δβγ but was purified using the same method as described above for SEC Fraction Analysis.

NEDD4-2-GFP and ENaC αβγ were mixed at varying ratios and analyzed by FSEC on a Waters Acquity Arc Bio HPLC in TBS + 0.5 mM DDM. Final NEDD4-2-GFP concentration was held constant at 71.2 nM, while ENaC αβγ was added in equal amounts, five-fold excess, or ten-fold excess, then 50 μL of these mixtures were injected onto a Superose 6 Increase 10/

300 column (Cytiva). The GFP channel was monitored to detect a shift in NEDD4-2-GFP elution position. The earlier-eluting peak that increases in height and area with increasing ENaC αβγ concentration elutes before both the ENaC αβγ alone and NEDD4-2-GFP alone peak positions in tryptophan fluorescence (not shown), indicating that the species is larger than either protein on its own.

Data were exported from Waters Empower and processed by Appia, then analyzed entirely in Appia Web.

## Supporting information

**S1 Video. A demonstration of the various features and modes of Appia Web.**
(MP4)

**S1 File. A previously-written script for analysis of incubation time and MOI effect on expression levels of a fluorescently tagged protein.**
(R)

**S1 Data. CSV table used to prepare Fig 4.**
(CSV)

**S2 Data. Analytic and preparative data used to prepare Fig 5.**
(ZIP)

**S3 Data. CSV table used to prepare Fig 6.**
(CSV)

## Acknowledgments

We are extremely thankful to the OHSU Medicinal Chemistry Core for providing sample Agilent data during development, and for helpful comments and discussion. Thanks also to Eric Gouaux, Steve Mansoor, and Shivani Ahuja for Shimadzu data. We would also like to thank the lively community of python, plotly, R, and ggplot users. Without the wealth of knowledge and help provided by these communities, Appia could never have been written. This paper would lose valuable biological perspective without the example chromatography data kindly provided by James Cahill and Alex Houser. Finally, we would like to thank Alex Houser, James Cahill, Kim Hartfield, Sigrid Noreng, Arpita Bharadwaj, for valuable feature suggestions and frequent bug reports.

## Data availability

Raw chromatography data are available in supplementary information. Appia source code is available at DOI: 10.5281/zenodo.6975032 under the MIT license. Appia is written in the following languages and packages under their respective licenses: python, PSF license; R, GPL-2 | GPL-3; tidyverse, MIT; Plotly, MIT. Copies of the relevant licenses are available in the Appia GitHub repository.

## Author Contributions

**Conceptualization:** Richard Posert.

**Funding acquisition:** Isabelle Baconguis.

**Methodology:** Richard Posert.

**Project administration:** Isabelle Baconguis.

**Software:** Richard Posert.

**Writing – original draft:** Richard Posert.

**Writing – review & editing:** Isabelle Baconguis.

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
