## [Decision Letter · Decision Letter 0]

6 Dec 2022

PONE-D-22-23933Appia: simpler chromatography analysis and visualizationPLOS ONE

Dear Dr. Isabelle Baconguis,

Thank you for submitting your manuscript to PLOS ONE. After careful consideration, we feel that it has merit but does not fully meet PLOS ONE’s publication criteria as it currently stands. Therefore, we invite you to submit a revised version of the manuscript that addresses the points raised during the review process. Please submit your revised manuscript by Jan 20 2023 11:59PM. If you will need more time than this to complete your revisions, please reply to this message or contact the journal office at plosone@plos.org. Please include the following items when submitting your revised manuscript:A rebuttal letter that responds to each point raised by the academic editor and reviewer(s). You should upload this letter as a separate file labeled 'Response to Reviewers'.A marked-up copy of your manuscript that highlights changes made to the original version. You should upload this as a separate file labeled 'Revised Manuscript with Track Changes'.An unmarked version of your revised paper without tracked changes. You should upload this as a separate file labeled 'Manuscript'.

We look forward to receiving your revised manuscript.

Kind regards,

Tommaso Lomonaco, Ph.D

Academic Editor

PLOS ONE

Journal Requirements:

This work was supported by the National Institute of General Medical Sciences (T32GM071338, initially to support RP, and R01GM138862 to IB). The content is solely the responsibility of the authors and does not necessarily represent the official views of the National Institutes of Health.

This work was supported by the National Institute of General Medical Sciences (T32GM071338, initially to support RP, and R01GM138862 to IB). The content is solely the responsibility of the authors and does not necessarily represent the official views of the National Institutes of Health.

Reviewers' comments:

Reviewer's Responses to Questions

**Comments to the Author**

1. Is the manuscript technically sound, and do the data support the conclusions?

Reviewer #1: Yes

Reviewer #2: Partly

2. Has the statistical analysis been performed appropriately and rigorously? 

Reviewer #1: Yes

Reviewer #2: N/A

3. Have the authors made all data underlying the findings in their manuscript fully available?

Reviewer #1: Yes

Reviewer #2: Yes

4. Is the manuscript presented in an intelligible fashion and written in standard English?

Reviewer #1: Yes

Reviewer #2: Yes

5. Review Comments to the Author

Reviewer #1: This work by the lab of Dr. Baconguis provided a useful tool to analyze HPLC data. The manuscript is well written, and the result presented is solid. Overall, I highly recommend this for publication.

Reviewer #2: Summary

Posert and Baconguis present the tool “Appia”, which is a set of scripts that allows users to convert chromatography data from major chromatography instruments such as Waters, Agilent, Shimadzu, and AKTA into a standardized format. The Appia suite of scripts allow users to set up an Appia web server for viewing the processed data in a browser via the Appia Web interface. The Appia Web interface provides data visualization through an interactive interface, which is self-explanatory and easy to use. Appia is proposed to help users overcome barriers in analysis of chromatography data both at the individual user level and between users in different laboratories.

In general, the manuscript is well-written, and presents a tool of interest to the broad readership of PLOS ONE. While Appia Web is explained in detail, more general information about Appia is lacking. Importantly, in our opinion, the first preprint on bioRxiv (submitted April 2022) answered several of the questions that we had after reading the submitted manuscript, both in terms of understanding how Appia works and what user-specific gaps it fills. As such, we recommend the authors to expand the current, main text with much of their previous version of the manuscript to unpack Appia and make it more approachable for the readership of the journal and potential users of the tool. We recommend publishing after major revisions according to our comments below.

Major comments to the manuscript

• Line 11-14: Rather than focusing on types of analysis that the current version of Appia doesn’t support (e.g. “integration of peak width, baseline subtraction”), the authors should consider highlighting the features that Appia does support in the abstract.

• The introduction of the manuscript should include a more thorough introduction to Appia than it currently does. Appia is only mentioned in the last sentence (line 49-50) and from this sentence is it not clear what Appia is.

• The authors introduce their tool as useful for various chromatographic applications but – since this is the main field of the authors – only provide examples for protein biochemistry applications. Nevertheless, it would further underline the strength, versatility, and accessibility of Appia, while also targeting the broad readership of PLOS ONE, if the authors included another, non-protein-based example in the manuscript.

• The authors describe that Appia converts chromatography data into a “tidy” format. When reading the manuscript, it is not clear what this “tidy” format entails and how it differs from the format provided by the manufacturers. The authors should give a brief outline of the structure of the “tidy” data in the text (line 53) and should consider showing the Appia data format compared to data in one of the manufacturers formats in an early figure, similar to the comparison between the traces in Figure 3. This would help readers and potential users to understand the importance of the software, in addition to equip them with the proper background to access the files and their content (and to detect potential flaws).

• Does the “tidy” format also contain the normalized data? From the provided data files in the Supporting information, it seems so, but the manuscript is missing information on the normalization; How it works and whether the user can choose to which data the data set is normalized (see line 75). The authors should provide more details on this step.

• Figure 1 doesn’t fit well with the current outline and scope of the manuscript. Visually, it is nice, as it highlights properties important in SEC analysis, but the figure is not central to understanding the work presented. It would be more relevant to include the workflow of Appia and how it is designed and looks like for the user in Figure 1. The authors could also consider including an outline of the “tidy” format in such a figure. For instance, the authors should consider referring to Figure 1 in line 64.

• A potential barrier in the analysis of chromatography data from different systems is the flow cell path length. It is not clear whether info about the flow cell path length is part of the “basic information” (line 60) that users are prompted for when starting an Appia Experiment. If not, the authors should consider adding it as a feature to further ease comparison of traces between different chromatography systems. Either way, the authors should explicitly state what info is asked for when starting an Appia “Experiment”.

• State-of-the-art in the field of chromatography data visualization and analysis is not clear from the current form of the manuscript, e.g. what do people usually do with this type of data (Gnuplot, GraphPad Prism, Origin, Excel?), and therefore the “proved advantage over existing alternatives” is not entirely clear (see PLOS ONE guidelines on software publications). Some of this is nicely described in the first preprint.

• In the demo Appia Web, the two FPLC examples are presented without axis legend (only on the hoover-over), neither is the axis legend added when the image is downloaded (if plotted alone). The authors should consider adding this information in the script if this is the case for all displayed FPLC data.

• In line 65, the authors write “… the data will be uploaded to the Appia Web database for viewing through a browser”. The authors should consider making it clear for the reader that the Appia database is local. Furthermore, the authors should elucidate more on the data storage and data protection within the Appia Web database, including how individual users may or may not access each other’s data within a single database. This is nicely expanded upon in the first preprint.

• In figure 4 and the corresponding text the authors explain how they use Appia Web to compare data from paired FSEC/SEC experiments. With Appia Web it should be possible to overlay the data generated by FSEC and SEC in one plot. The authors should provide such an overlay plot for the data shown in figure 4. If it is not possible to create such an overlay plot yet, it would be helpful to implement such an option for easy comparison. Elaborate how this is better than the manufacturers solutions.

• It is not clear which proteins used in the experiments outlined in Figure 4 and 5. The authors should include which protein were used here, particularly as it is difficult to follow exactly which EnACs were used when: In lines 187 and 205, is it the same EnAC or are different EnACs (αβγ vs δβγ) used?

• The authors should add something about the outlook and long-term utility of Appia. What is the future growth of the software? How could it be further improved? Could integration of area under the curve/peak and Gaussian functions of polydisperse peaks be included in the software or does this lie within the domain of more advanced forms of chromatography? What about the maintenance/future of the software? What is the likelihood for it to become outdated within a couple of years of installation? Users should have somewhat an insurance that the software will be taken good care of in the future if they are to include it in their workflow.

• The manuscript is missing a conclusion.

Comments to the software

• The authors provide few details on how to run Appia in Python. The GitHub repository is lacking details here as well. To make Appia available for a broad range of users more details should be given in the manuscript as well as in the GitHub repository on how to run Appia in Python. According to the GitHub page, the authors previously tested out a GUI for Appia, but decided against it. The authors should reconsider whether the addition of a GUI would be helpful for users to whom command line work is perceived as difficult/a bottleneck.

• Along the lines above, the authors should consider adding a -h function to help new users run appia.py in the command line.

• It would be nice to have more options for export formats in Appia Web than just .png. The authors should consider adding other options such as .svg, .pdf, .tiff. Furthermore, it seems like Appia Web only allows the user to download images. The authors should consider allowing users to download the generated .csv data to facilitate the exchange of the converted data (and the transparency for the user themselves).

Minor comments

• The readability of the manuscript would greatly improve if the authors gave some of the sections in the Results section headers.

• Line 20: “Foundational” is a word, but it reads odd. In this context, the word “fundamental” would read better.

• Line 31” “are” is missing between analyte and sufficient.

• Line 33: Gaussian should be written with uppercase: Gaussian.

• Line 36: “If two” should be “Two”.

• Line 37: New section, so “these” should be named “the position, heights and width”.

• Line 39: “Our field” should be “the field”.

• Line 42: The sentence with “complex chromatographic assays” reads odd and should be rephrased.

• Line 41 and 43: “Chromatogram manufacturer” must be “chromatographic system manufacturer”?

• A full stop is missing for the sentences that end in lines 36 and 101.

• Line 92: The authors should consider adding info detailing that their protein of interest is fused to a fluorescent protein, and which tag they use, e.g. “… a common assay in our lab is exploration of expression levels of a given protein construct fused to green fluorescent protein (GFP).”

• Line 94: The abbreviation MOI (multiplicity of infection) should be explained in the main text, and not only presented in the Experimental procedures (line 176).

• Line 107: It says “… with a clear plot layout enabled by Appia”. Surely, this was not only possible due to Appia as other custom scripts could have enabled it as well.

• Line 111: fluorescence-detection size-exclusion chromatography (FSEC) should to be introduced the first time it appears.

• Line 119: Experiments should be written in all lower case.

• In lines 168 and 221 it should be repository instead of repo.

• In Appia Web, the buttons to the left always display “HPLC” even though FPLC data are loaded. The authors should consider giving the buttons a uniform format accordingly.

6. PLOS authors have the option to publish the peer review history of their article (what does this mean?). If published, this will include your full peer review and any attached files.

Reviewer #1: No

Reviewer #2: **Yes: **Henriette Elisabeth Autzen

---

## [Author Response · Author response to Decision Letter 0]

17 Dec 2022

Response to Reviewers

We thank the reviewers for their thorough and helpful comments. We believe that both the manuscript and the software are stronger as a result. Below we respond point-by-point. We will use line numbers from the “manuscript.docx” file (with changes accepted) throughout this reply.

Major comments to the manuscript

• Line 11-14: Rather than focusing on types of analysis that the current version of Appia doesn’t support (e.g. “integration of peak width, baseline subtraction”), the authors should consider highlighting the features that Appia does support in the abstract.

We appreciate this suggestion and have modified the abstract to focus instead on the advantages of portability that Appia provides.

• The introduction of the manuscript should include a more thorough introduction to Appia than it currently does. Appia is only mentioned in the last sentence (line 49-50) and from this sentence is it not clear what Appia is.

We have added a paragraph to the end of the introduction to correct this oversight (lines 58—67).

• The authors introduce their tool as useful for various chromatographic applications but – since this is the main field of the authors – only provide examples for protein biochemistry applications. Nevertheless, it would further underline the strength, versatility, and accessibility of Appia, while also targeting the broad readership of PLOS ONE, if the authors included another, non-protein-based example in the manuscript.

We agree, but unfortunately the only non-protein data we have access to is from a core facility, and so we cannot share precise details about the experiments. We have, however, added a general description of Appia’s use in medicinal chemistry to the Results (lines 166—170).

• The authors describe that Appia converts chromatography data into a “tidy” format. When reading the manuscript, it is not clear what this “tidy” format entails and how it differs from the format provided by the manufacturers. The authors should give a brief outline of the structure of the “tidy” data in the text (line 53) and should consider showing the Appia data format compared to data in one of the manufacturers formats in an early figure, similar to the comparison between the traces in Figure 3. This would help readers and potential users to understand the importance of the software, in addition to equip them with the proper background to access the files and their content (and to detect potential flaws).

We agree that our description of “tidy” data was lacking. We have added a more detailed explanation (lines 87—91) and a figure (Fig 2) to clarify what “tidy” data is. We believe that addition of manufacturer data examples may confuse the reader, since they are all different and difficult to read --- one of the main initial motivations behind the development of Appia.

• Does the “tidy” format also contain the normalized data? From the provided data files in the Supporting information, it seems so, but the manuscript is missing information on the normalization; How it works and whether the user can choose to which data the data set is normalized (see line 75). The authors should provide more details on this step.

We have provided more details on the normalization step (lines 119—124) and added a figure demonstrating “normalization” of an example set of traces (Fig 3). 

• Figure 1 doesn’t fit well with the current outline and scope of the manuscript. Visually, it is nice, as it highlights properties important in SEC analysis, but the figure is not central to understanding the work presented. It would be more relevant to include the workflow of Appia and how it is designed and looks like for the user in Figure 1. The authors could also consider including an outline of the “tidy” format in such a figure. For instance, the authors should consider referring to Figure 1 in line 64.

We appreciate the reviewer’s compliment of the figure’s visuals, and agree that it does not fit very well in the flow of the document. We have removed it and replaced it with another figure from the manuscript which we believe operates as an acceptable overview of Appia’s functionality (new Fig 1).

• A potential barrier in the analysis of chromatography data from different systems is the flow cell path length. It is not clear whether info about the flow cell path length is part of the “basic information” (line 60) that users are prompted for when starting an Appia Experiment. If not, the authors should consider adding it as a feature to further ease comparison of traces between different chromatography systems. Either way, the authors should explicitly state what info is asked for when starting an Appia “Experiment”.

We have added a description of the minimal and optional “basic information” required to add an experiment to Appia (lines 76--81). We agree that flow cell path length is an important parameter in all forms of spectroscopy, and we have added a processing-time option to linearly scale all values by a user-defined factor (`appia process --scale-hplc`, see lines 81—83). This could be used to accomplish the desired effect of scaling results to have the same effective path length.

However, users must also be careful to inspect the readout provided by different manufacturers. Some (GE/Cytiva) report mAU directly, and so a simple path length scaling would be sufficient. On the other hand, some manufacturers (Shimadzu) report much larger values for the current through the detector. These would have to be scaled by some empirical value, and this value would likely be different for different sensitivity and gain settings. For this reason, we believe that direct numeric (as opposed to relative) comparison of values from different manufacturers is currently outside the scope of Appia, and did not include the scaling/path length value as required information. If this type of analysis is essential for some workflows, we are happy to begin work on it in as an issue in our GitHub repository.

• State-of-the-art in the field of chromatography data visualization and analysis is not clear from the current form of the manuscript, e.g. what do people usually do with this type of data (Gnuplot, GraphPad Prism, Origin, Excel?), and therefore the “proved advantage over existing alternatives” is not entirely clear (see PLOS ONE guidelines on software publications). Some of this is nicely described in the first preprint.

We agree that this information from the earlier preprint was valuable and have added some of it to the current manuscript (lines 215—223).

• In the demo Appia Web, the two FPLC examples are presented without axis legend (only on the hoover-over), neither is the axis legend added when the image is downloaded (if plotted alone). The authors should consider adding this information in the script if this is the case for all displayed FPLC data.

We appreciate this bug report. Axis labels have been added to FPLC data.

• In line 65, the authors write “… the data will be uploaded to the Appia Web database for viewing through a browser”. The authors should consider making it clear for the reader that the Appia database is local. Furthermore, the authors should elucidate more on the data storage and data protection within the Appia Web database, including how individual users may or may not access each other’s data within a single database. This is nicely expanded upon in the first preprint.

This is an important point --- we have added several notes about local hosting to the manuscript (lines 59, 62, and 103).

• In figure 4 and the corresponding text the authors explain how they use Appia Web to compare data from paired FSEC/SEC experiments. With Appia Web it should be possible to overlay the data generated by FSEC and SEC in one plot. The authors should provide such an overlay plot for the data shown in figure 4. If it is not possible to create such an overlay plot yet, it would be helpful to implement such an option for easy comparison. Elaborate how this is better than the manufacturers solutions.

We have added an FPLC overlay feature to Appia Web.

• It is not clear which proteins used in the experiments outlined in Figure 4 and 5. The authors should include which protein were used here, particularly as it is difficult to follow exactly which EnACs were used when: In lines 187 and 205, is it the same EnAC or are different EnACs (αβγ vs δβγ) used?

We have added text clarifying which ENaC heterotrimer is used in which experiment.

• The authors should add something about the outlook and long-term utility of Appia. What is the future growth of the software? How could it be further improved? Could integration of area under the curve/peak and Gaussian functions of polydisperse peaks be included in the software or does this lie within the domain of more advanced forms of chromatography? What about the maintenance/future of the software? What is the likelihood for it to become outdated within a couple of years of installation? Users should have somewhat an insurance that the software will be taken good care of in the future if they are to include it in their workflow.

We agree this is always an important point with software, and it’s a main reason why Appia uses simple CSV files. Even if the software stops working, data will always be accessible. However, our lab uses Appia almost exclusively to analyze chromatography data, and so we plan to support it indefinitely. We have added a note to this effect (lines 242—247).

• The manuscript is missing a conclusion.

We have corrected this oversight and added a conclusion.

Comments to the software

All of the software changes requested are visible in the updated web demo, at the same URL as in the text. We have split the first comment into two, since we believe the comment covers two distinct topics.

• The authors provide few details on how to run Appia in Python. The GitHub repository is lacking details here as well. To make Appia available for a broad range of users more details should be given in the manuscript as well as in the GitHub repository on how to run Appia in Python.

We apologize for lack of clarity on actually running Appia. We have added a few notes to both the text (lines 256—259) and the GitHub page (headings Installation, HPLC Processing, and FPLC Processing) to correct this oversight. We have also added more detail for the installation of Appia Web (lines 261—272 and GitHub heading Installation).

• According to the GitHub page, the authors previously tested out a GUI for Appia, but decided against it. The authors should reconsider whether the addition of a GUI would be helpful for users to whom command line work is perceived as difficult/a bottleneck.

We found that users rarely opened the GUI, instead using the bundled batch scripts to process their chromatography data. The GUI requires several more dependencies, making installation more challenging and impossible on some systems. For these reasons, we removed it. However, we agree that many users find CLI programs intimidating. We will consider adding a more versatile GUI in the future, or perhaps a more-comprehensive automated processing suite. 

• Along the lines above, the authors should consider adding a -h function to help new users run appia.py in the command line.

Appia already contains extensive help functionality in the CLI. We reproduce it at the end of this document.

• It would be nice to have more options for export formats in Appia Web than just .png. The authors should consider adding other options such as .svg, .pdf, .tiff. Furthermore, it seems like Appia Web only allows the user to download images. The authors should consider allowing users to download the generated .csv data to facilitate the exchange of the converted data (and the transparency for the user themselves).

We agree and have added a selector to the web interface for all options available in the plot building library we use: png, svg, jpg, and webp.

We also agree that .pdf and .tiff files are also important for use in publications and are exploring ways to add support for those filetypes to the web interface. Appia does come packaged with R scripts which can generate the requested filetypes as well as many others, but those require a familiarity with R and ggplot2 and so are not the best solution for all users.

We have opened an issue on this request in our GitHub repository and are investigating three potential solutions.

Minor comments

• The readability of the manuscript would greatly improve if the authors gave some of the sections in the Results section headers.

• Line 20: “Foundational” is a word, but it reads odd. In this context, the word “fundamental” would read better.

• Line 31” “are” is missing between analyte and sufficient.

• Line 33: Gaussian should be written with uppercase: Gaussian.

• Line 36: “If two” should be “Two”.

• Line 37: New section, so “these” should be named “the position, heights and width”.

• Line 39: “Our field” should be “the field”.

• Line 42: The sentence with “complex chromatographic assays” reads odd and should be rephrased.

• Line 41 and 43: “Chromatogram manufacturer” must be “chromatographic system manufacturer”?

• A full stop is missing for the sentences that end in lines 36 and 101.

• Line 92: The authors should consider adding info detailing that their protein of interest is fused to a fluorescent protein, and which tag they use, e.g. “… a common assay in our lab is exploration of expression levels of a given protein construct fused to green fluorescent protein (GFP).”

• Line 94: The abbreviation MOI (multiplicity of infection) should be explained in the main text, and not only presented in the Experimental procedures (line 176).

• Line 107: It says “… with a clear plot layout enabled by Appia”. Surely, this was not only possible due to Appia as other custom scripts could have enabled it as well.

• Line 111: fluorescence-detection size-exclusion chromatography (FSEC) should to be introduced the first time it appears.

• Line 119: Experiments should be written in all lower case.

• In lines 168 and 221 it should be repository instead of repo.

We appreciate all of the comments and have made the appropriate changes in the manuscript.

• In Appia Web, the buttons to the left always display “HPLC” even though FPLC data are loaded. The authors should consider giving the buttons a uniform format accordingly.

Appia web is now responsive to what types of data are being presented. HPLC/FPLC options will only be visible when the appropriate data is displayed.

Appia CLI Help

Appia help

$ appia --help 

usage: appia [-h] [-q | -v | --debug] {process,database,utils} ...

Process chromatography data and visualize it on the web.

positional arguments:

 {process,database,utils}

 process Process data

 database Manage CouchDB

 utils Utilities

options:

 -h, --help show this help message and exit

Verbosity:

 -q, --quiet Print Errors only

 -v, --verbose Print Info, Warnings, and Errors. Default state.

 --debug Print debug output.

Processing help

$ appia process --help 

usage: appia process [-h] [-i ID] [-o OUTPUT_DIR] [-k] [-c [COPY_MANUAL]] [-s [POST_TO_SLACK]]

 [--hplc-flow-rate HPLC_FLOW_RATE] [--fplc-cv FPLC_CV]

 [-n NORMALIZE NORMALIZE] [--strict-normalize]

 [--channel-mapping CHANNEL_MAPPING [CHANNEL_MAPPING ...]] [-r REDUCE]

 [-d [CONFIG]] [--overwrite] [-p] [-f FRACTIONS [FRACTIONS ...]] [-m ML ML]

 files [files ...]

positional arguments:

 files Glob or globs to find data files. For instance, "traces/*.arw"

options:

 -h, --help show this help message and exit

File IO:

 -i ID, --id ID Experiment ID. Default to name of HPLC Sample Set (Waters over Shimadzu, if present) or FPLC file name.

 -o OUTPUT_DIR, --output-dir OUTPUT_DIR

 Directory in which to save CSVs and plots. Default makes a new dir with experiment name.

 -k, --no-move Process data files in place (do not move to new directory)

 -c [COPY_MANUAL], --copy-manual [COPY_MANUAL]

 Copy R template file for manual plot editing. Argument is directory relative to Appia root in which templates reside.

 -s [POST_TO_SLACK], --post-to-slack [POST_TO_SLACK]

 Send completed plots to Slack. Need a config JSON with slack token and channel.

Processing Options:

 --hplc-flow-rate HPLC_FLOW_RATE

 Manually override flow rate. Provide a single number in mL/min

 --fplc-cv FPLC_CV Column volume for FPLC data. Default is 24 mL (GE/Cytiva 10/300 column).

 -n NORMALIZE NORMALIZE, --normalize NORMALIZE NORMALIZE

 Set maximum of this range (in mL) to 1

 --strict-normalize Also set minimum of normalization range to 0

 --channel-mapping CHANNEL_MAPPING [CHANNEL_MAPPING ...]

 Channel mappings for old Shimadzu instruments. Default: A Trp B GFP

 --scale-hplc SCALE_HPLC

 Scale signal values by a factor. For instance, --scale-hplc 0.5 will reduce all

 signal values by 1/2. Could be used to compare instruments with different flow cell path lengths.

Web Upload:

 -r REDUCE, --reduce REDUCE

 Reduce web HPLC data points to this many total. Default 1000. CSV files are saved at full temporal resolution regardless.

 -d [CONFIG], --database [CONFIG]

 Upload experiment to couchdb. Optionally, provide config file location. Default config location is "config.json" in appia directory. Enter "env" to use environment variables instead.

 --overwrite Overwrite database copy of experiment with same name without asking

Auto Plots:

 -p, --plots Make default plots

 -f FRACTIONS [FRACTIONS ...], --fractions FRACTIONS [FRACTIONS ...]

 SEC fractions to fill in. Default is none. Giving two numbers fills inclusive range; a third sets interval. E.g., 2 10 2 fills even fractions between 2 and 10.

 -m ML ML, --ml ML ML Inclusive range for auto-plot x-axis, in mL. Default is 5 to 25. To auto-set one limit, type `auto` instead of a number.

Database management help

$ appia database –help

usage: appia database [-h] [-l] [-d DELETE [DELETE ...]] [-i INSPECT [INSPECT ...]]

 [--download DOWNLOAD [DOWNLOAD ...]] [--check-versions] [--migrate]

 config

positional arguments:

 config Config JSON file

options:

 -h, --help show this help message and exit

 -l, --list Print list of all experiments in database

 -d DELETE [DELETE ...], --delete DELETE [DELETE ...]

 Delete experiment(s) by name

 -i INSPECT [INSPECT ...], --inspect INSPECT [INSPECT ...]

 Print information about experiments

 --download DOWNLOAD [DOWNLOAD ...]

 Save experiments from the database as a .csv. Note that these may have been downsampled.

 --check-versions List experiments, categorized by version.

 --migrate Download and upload all experiments to migrate them to a new version. Back up first!!!

Utilities help

$ appia utils –help

usage: appia utils [-h] [-c [COPY_MANUAL]]

options:

 -h, --help show this help message and exit

 -c [COPY_MANUAL], --copy-manual [COPY_MANUAL]

 Copy R manual plotting template. Argument is directory in which template resides, relative to Appia root.

---

## [Decision Letter · Decision Letter 1]

27 Dec 2022

Appia: simpler chromatography analysis and visualization

PONE-D-22-23933R1

Dear Dr. Isabelle Baconguis,

We’re pleased to inform you that your manuscript has been judged scientifically suitable for publication and will be formally accepted for publication once it meets all outstanding technical requirements.

Kind regards,

Tommaso Lomonaco, Ph.D

Academic Editor

PLOS ONE

Reviewers' comments:

Reviewer's Responses to Questions

**Comments to the Author**

1. If the authors have adequately addressed your comments raised in a previous round of review and you feel that this manuscript is now acceptable for publication, you may indicate that here to bypass the “Comments to the Author” section, enter your conflict of interest statement in the “Confidential to Editor” section, and submit your "Accept" recommendation.

Reviewer #1: All comments have been addressed

Reviewer #2: All comments have been addressed

2. Is the manuscript technically sound, and do the data support the conclusions?

Reviewer #1: Yes

Reviewer #2: Yes

3. Has the statistical analysis been performed appropriately and rigorously? 

Reviewer #1: Yes

Reviewer #2: N/A

4. Have the authors made all data underlying the findings in their manuscript fully available?

Reviewer #1: Yes

Reviewer #2: Yes

5. Is the manuscript presented in an intelligible fashion and written in standard English?

Reviewer #1: Yes

Reviewer #2: Yes

6. Review Comments to the Author

Reviewer #1: The authors properly addressed all concerns. The manuscript should be accepted for publication in plos one.

Reviewer #2: The authors have answered all of our questions and considered all of our comments to great satisfaction. The resulting re-submitted manuscript reads well and, in our opinion, makes the tool Appia much more approachable for the readership of the journal and potential users than the original submission. In addition, we are happy that the Github description has been expanded upon as well. We recommend accepting the revised manuscript. Below are a few minor comments and kind suggestions to the text that the authors might consider incorporating when editing the final version. The lines refer to the resubmitted manuscript.

• Line 72: Consider changing “any supported chromatogram’s data” into “a range of chromatographic data”.

• Line 74: Should “into” be “as”?

• Line 76: that can be omitted and “raw” could be added in front of data files.

• Line 80-81: Consider adding “at a” after “or one”

• Line 89: Consider substituting perhaps with e.g.,

• Line 92: Consider adding s to program and package

• Line 92-93: Consider omitting the brackets, and add “,” and “;”

• Line 103-104: Consider replacing “ -we (the authors)" with “, and the authors and developers of Appia neither” after hosted

• Line 108: Consider removing “if” and adding “who” after users

• Line 114: Consider adding “, and preparative data” after analytic data and changing the following to “as they typically require different visualization”.

• Line 189-190: Consider adding “between proteins and ligands” after interactions, “;” after (Fig. 6), and “of the ligand-bound protein peak” before to the unbound peak.

• Line 194: Consider removing “plainly” and add “in Fig. 6” after “traces”.

• Line 228-229: Consider rephasing, “While Appia is unable to run… nor process…, Appia provides a free…”

7. PLOS authors have the option to publish the peer review history of their article (what does this mean?). If published, this will include your full peer review and any attached files.

Reviewer #1: No

Reviewer #2: **Yes: **Henriette Elisabeth Autzen

---

## [Editor Report · Acceptance letter]

2 Jan 2023

PONE-D-22-23933R1 

Appia: simpler chromatography analysis and visualization 

Dear Dr. Baconguis:

I'm pleased to inform you that your manuscript has been deemed suitable for publication in PLOS ONE. Congratulations! Your manuscript is now with our production department. 

Kind regards, 

on behalf of

Dr. Tommaso Lomonaco 

Academic Editor

PLOS ONE